# Aberrant Effective Connectivity Within and Between the Default Mode, Executive Control, and Salience Networks in Chronic Insomnia Disorder—Toward Identifying the Hyperarousal State

**DOI:** 10.3390/biomedicines13061293

**Published:** 2025-05-24

**Authors:** Todor Georgiev, Rositsa Paunova, Anna Todeva-Radneva, Krasimir Avramov, Aneliya Draganova, Sevdalina Kandilarova, Kiril Terziyski

**Affiliations:** 1Department of Pathophysiology, Medical University of Plovdiv, 4002 Plovdiv, Bulgaria; kavramov@pathophysiology.info (K.A.); adraganova@pathophysiology.info (A.D.); kterziyski@pathophysiology.info (K.T.); 2Department of Psychiatry and Medical Psychology, Medical University of Plovdiv, 4002 Plovdiv, Bulgaria; rositsa.paunova@mu-plovdiv.bg (R.P.); anna.todeva@mu-plovdiv.bg (A.T.-R.); sevdalina.kandilarova@mu-plovdiv.bg (S.K.); 3Translational and Computation Neuroscience Group, Research Institute and SRIPD-MUP, 4002 Plovdiv, Bulgaria

**Keywords:** chronic insomnia, hyperarousal, pathogenesis, resting-state functional MRI, effective connectivity, dynamic causal modeling, default mode network, salience network, dorsolateral prefrontal cortex, anterior insula

## Abstract

**Background**: Chronic insomnia (CID) is a highly prevalent sleep disorder, yet the precise mechanisms underlying it remain incompletely understood. The aim of this study is to analyze effective connectivity between key regions of the default mode network (DMN), executive control network (ECN), and salience network (SN) in patients with CID as potential neurologic correlates of the hyperarousal state. **Methods**: Thirty-one CID patients and 24 healthy controls (HC) were recruited. All the subjects filled out the Insomnia severity index scale (ISI), Beck depression inventory (BDI), and Epworth sleepiness scale (ESS), underwent polysomnography, and were scanned on functional magnetic resonance imaging. Statistical Parametric Mapping 12 was used to analyze the results. Spectral dynamic causal modeling was applied to the chosen regions of interest. **Results**: There were three significant connections present in the CID group—inhibitory from the dorsolateral prefrontal cortex (DLPFC) to the right hippocampus (Hippocamp R); excitatory from the dorsomedial prefrontal cortex to the ventromedial prefrontal cortex; and excitatory from the common medial prefrontal cortex to the right anterior insula (AIR). Two statistically significant excitatory connections were lacking in the patients’ group—from the posterior cingulate cortex (PCC) to AIR, and from precuneus to PCC. CID patients scored higher on the ISI and BDI. Significant negative correlations between DLPFC-Hippocamp R connectivity and both ISI and BDI scores were identified. **Conclusions**: Disruptions within the DMN and between the DMN, SN, and ECN reflect an impaired ability to appropriately shift between internally and externally directed cognitive states—an imbalance that potentially underlies the hyperarousal state of CID.

## 1. Introduction

Chronic insomnia disorder (CID), identified based on established diagnostic criteria, is estimated to impact approximately 5% to 15% of adults [1,2,3]. Despite its high prevalence, the precise pathogenic mechanisms underlying this disorder remain partially understood [4]. A key conceptual model for the pathogenesis of CID is the hyperarousal hypothesis, which is based on the observed persistent state of physiological, cognitive, and emotional hyperactivation in CID patients [4,5,6]. The model posits that individuals with chronic insomnia exhibit a heightened arousal, which is not limited to the nighttime but extends throughout the wake state, implicating dysregulation of neuroendocrine and functional brain circuits responsible for sleep-wake control and emotional regulation [7]. These changes include overactivated hypothalamic-pituitary axis and electroencephalographic (EEG) findings of heightened high-frequency activity during sleep in CID patients, indicative of cortical overactivation [4,8]. Several other facets of the hyperarousal have been described, such as elevated metabolic activity, altered cognitive domain, dysregulated stress response, and altered synchronization of cortical networks, with no clear evidence of structural lesions in the brain [6,9,10]. However, while this conceptual model is clinically informative, it does not provide an insight into the precise neural correlates underlying CID [11]. Structural imaging studies fail to adequately identify the neural structures involved in the pathogenesis of CID and show numerous and inconsistent alterations across regions of the gray matter in the central nervous system, mainly involving the hippocampus and cortical gray matter [12,13].

Resting-state functional MRI (rs-fMRI) is more informative in identifying the neurobiological underpinnings of CID, as it captures the brain’s intrinsic activity in the absence of external attention-demanding tasks and stimulations [14]. Albeit bearing a resemblance to the transition into sleep and being informative regarding spontaneous brain dynamics, rs-fMRI does not replicate sleep and should not be considered as an analog of sleep initiation [15]

Accumulating neuroimaging evidence points to dysregulated functional connectivity (FC) within and between large-scale brain networks at rest as a key mechanism in the pathogenesis of CID and its characteristic hyperarousal state [16,17,18]. Specifically, three major brain networks are implicated in the regulation of sleep-wake states: the default mode network (DMN); the executive control network (ECN); and the salience network (SN). These three networks are critical for self-referential thinking, internal mentation, attentional control, emotional regulation, and cognitive flexibility [19,20,21]. All the aforementioned elements have been integrated into the neurocognitive model at various stages throughout the evolution of our comprehensive understanding of the psycho-behavioral manifestations of insomnia.

The DMN comprises regions such as the medial prefrontal cortex (MPFC), posterior cingulate cortex (PCC), and precuneus; the SN includes the anterior insula and anterior cingulate cortex, while the ECN encompasses the dorsal prefrontal cortex (DLPFC) and posterior parietal cortex (PPC) [21,22,23]. Functional connectivity studies in CID patients have consistently shown disrupted synchronization within the DMN, reduced coupling between the MPFC and limbic structures and aberrant SN engagement during both rest and cognitive tasks [24,25,26,27]. Altered FC between the insula—a central hub of the SN—and nodes of the DMN has also been observed, with decreased connectivity correlating positively with anxiety symptoms, which further supports the link between emotional dysregulation and the hypervigilant state in CID [25].

In addition, hypoactivation of the MPFC during cognitive tasks suggests impairments in self-referential processing and difficulties in cognitive disengagement, while reduced FC between the MPFC and PCC has been associated with memory consolidation deficits and broader cognitive dysfunction in individuals with insomnia [28,29].

Moreover, disruptions in ECN are a common finding among patients with impaired sleep. Reduced FC in the DLPFC and other ECN regions has been linked to impaired cognitive control and increased vulnerability to emotional dysregulation among patients with cognitive impairment and insomnia symptoms [30]. In addition, reduced dynamic FC between anterior insula and DLPFC (major nodes of the anterior SN and ECN) is observed in insomniacs, attributing to the impaired SN dynamics, which fails to modulate the enhanced top-down cognitive control [31,32,33].

However, while FC provides an insight into which regions are co-active, little is known about the causal connectivity changes that may underlie these disturbances and contribute to the persistent hyperarousal state in CID. Effective connectivity (EC) describes the influence of one brain region over another, incorporating temporal and causal effects of the regions in a neural circuit [34]. Currently, only one study has explored the EC between the main brain networks in patients with CID. Li et al. report aberrant EC between the right anterior insula (AIR), a key hub in the SN, and the precuneus (a major region of the DMN) [35]. These alterations are related to cognitive impairment, altered working memory, and decision making, which are characteristic traits of CID patients, but do not provide insights into the observed hypervigilant state of CID.

Based on the findings in the available literature, we hypothesized that EC between the major hubs of the DMN, ECN, and SN may be disrupted in individuals with CID, impairing the SN’s ability to modulate and “switch” between DMN and ECN. Motivated by the lack of sufficient research on the EC in CID and wishing to fill this gap, we aimed to explore the EC in patients with CID compared to healthy individuals using Dynamic Casual Modeling. In addition, we were interested in the possible correlation between brain connectivity measures and clinical features of the patients.

## 2. Materials and Methods

This prospective, cross-sectional study was performed at the Medical University of Plovdiv, Bulgaria, and approved by the ethics committee of the university (grant number: HO-11/2021 (P8540/2021). Written informed consent was obtained from all participants. Thirty-one CID patients (mean age 33.8 ± 9.1; nine males), complying with the diagnostic criteria of the “International classification of sleep disorders, III-rd. edition, TR” (ICSD-III TR), were recruited for this study, as well as 24 age- and sex-matched HC (mean age 29.5 ± 7; nine males).

The inclusion criteria were: (a) age between 18 and 65 years; and (b) diagnosis of CID, according to the ICSD-III TR criteria (for the patients’ group). The following exclusion criteria were applied to both groups: (a) another sleep disorder, such as moderate or severe sleep apnea, restless legs syndrome, periodic limb movement disorder, impaired circadian rhythmicity, etc.; (b) psychiatric disorder; (c) intake of psychoactive medications; (d) counterindications for MRI scan; (e) shift work; (f) severe somatic diseases, significantly worsening sleep quality and quantity; and (g) neurologic diseases. All the participants met these criteria and were included in the study.

*Sleep and emotion questionnaires.* All the participants filled out the following scales, assessing insomnia symptoms, sleep quality and depressive traits: the Insomnia severity index (ISI); the Beck depression inventory (BDI); and the Epworth sleepiness scale (ESS).

*Polysomnography*. All the subjects were assessed by clinical interview and underwent unattended, single-night, home-based polysomnography (PSG) using NOX A1 PSG systems, Reykjavík, Iceland, to rule out a concomitant sleep disorder. The results were manually scored by two certified somnologists (K.T. and A.D.) based on version 2.3 of the American Academy of Sleep Medicine criteria. The main parameters include reported sleep onset latency (SOL), total sleep time (TST), time in bed (TIB), sleep efficiency (SE), sleep stages (N1, N2, N3 and REM) in minutes and percentage, total wake time, wake after sleep onset (WASO), and REM latency.

*MRI scanning and Data Analysis*. The scanning of the participants was performed on a 3-T MRI system (GE Discovery 750 w) and included a high-resolution structural scan (Sag 3D T1 FSPGR, slice thickness 1 mm, matrix 256 × 256, TR (relaxation time)—7.2 ms, TE (echo time)—2.3, flip angle 12°), and a functional scan 2D Echo Planar Imaging (EPI), slice thickness 3 mm, 36 slices, matrix 64 × 64, TR—2000 ms, TE—30 ms, flip angle 90°, 192 volumes. Before the EPI sequence, subjects were instructed to remain as still as possible with their eyes closed and not to think about anything in particular.

Data analysis was conducted using SPM12 (Statistical Parametric Mapping; http://www.fil.ion.ucl.ac.uk/spm/; access date 22 January 2022) implemented in MATLAB R2020b (Windows version). During the preprocessing stage of the functional images, standard procedures were applied: realignment; co-registration with structural scans; spatial normalization to the Montreal Neurological Institute (MNI) template; and smoothing using a Gaussian kernel with a full width at half maximum of 6 mm.

Resting-state data were analyzed at the first level using a general linear model (GLM) applied to the time series. Regions of interest (ROIs), defined as 6-mm radius spheres, were selected based on the literature review of numerous neuroimaging studies that identified these areas as key nodes within the SN, DMN, and ECN. For this study, we tested three effective connectivity models, containing 6 to 8 ROIs. The number of ROIs within each model was tailored to the limitations of the software we used. The first model aimed to explore the between-group differences of the connectivity strengths among hubs of the DMN. Subsequently, with the second model we explored the relationship among major regions of the DMN and SN. Finally, we tested a model that included the key regions of DMN, SN, and ECN in order to determine possible effective connectivity aberrations among the three large-scale brain networks that may be pivotal for the pathophysiology of CID. The ROIs forging the within- and between-group statistically significant connections (based on the Kolmogorov–Smirnov and Mann–Whitney tests) and their MNI coordinates are presented in Table 1. Further details of all the models are provided in the Appendix A.

Subsequently, spectral dynamic causal modeling (spDCM) was applied to the chosen ROIs. A fully connected model was implemented, whereby each region was assumed to influence all others. Unlike stochastic DCM, spectral DCM estimates EC based on the cross-spectral density of neuronal state fluctuations rather than from raw time series data. The individual spDCM models were jointly estimated using the parametric empirical Bayes (PEB) approach available in SPM12. Finally, the resulting A-matrix parameters—representing the EC strengths—were extracted and used for further statistical analyses in SPSS.

*Statistical analysis*. Statistical analysis of all the demographic and clinical data, as well as connectivity strengths and PSG data, were analyzed using SPSS 25.0 (IBM Corp., Armonk, NY, USA) for Windows. Normality of distribution was tested using the Kolmogorov–Smirnov test. Demographic characteristics, questionnaire results, and sleep parameters were compared using the Kolmogorov–Smirnov test, or chi-square tests. Correlations were performed using Pearson’s correlation method. One sample test against zero was used to identify the statistically significant connections within each group. The Mann–Whitney U test was applied to search for differences between groups. Given the number of comparisons, the two-stage Benjamini–Hochberg procedure was applied to control the false discovery rate (FDR), with a significance threshold set at q < 0.05.

## 3. Results

### 3.1. Demographic and Clinical Characteristics

No significant differences were observed between patients and HC in age, sex, and education level. CID patients scored higher on the ISI and BDI. As expected, CID patients did not show higher levels of sleepiness compared to HC, despite the reported sleep complaints. The CID group demonstrated lower levels of self-reported sleep quality and had significantly higher WASO and total time in bed compared to the HC group. PSG data showed a higher N2 stage and lower N3 stage in minutes and as a percentage of total sleep time, compared to good sleepers. The demographic and clinical data are presented in Table 2.

### 3.2. One-Sample Kolmogorov–Smirnov Test Results

The one-sample test identified connections that were significantly different from zero in each group. We observed numerical hypoconnectivity (fewer significant connections) within and among the DMN, SN, and ECN, as well as a reduced number of self-inhibitory connections in the patient group. Concomitantly, the significant connections within and between the three major brain networks were greater in the HC group. For a detailed description of the effective connectivity on a group level, see Appendix A.

### 3.3. Two-Sample Mann–Whitney Test Results

The test yielded 5 connections with statistically significant differences of the connectivity strengths between the HC and CID groups, namely: DMPFC → VMPFC (excitatory); MPFC → AIR (excitatory); DLPFC → Hippocamp R (inhibitory); PCC → AIR (excitatory); and Precuneus → PCC (excitatory). In addition, the first three connections were only present in the patient group, whereas the excitatory PCC → AIR and Precuneus → PCC connections were only significant in the HC group (Table 3 and Figure 1). Following two-stage false discovery rate (FDR) correction, four out of the five connections remained statistically significant (q < 0.05): DMPFC–VMPFC (*p* = 0.049, q = 0.049), MPFC–AIR (*p* = 0.014, q = 0.028), DLPFC–Hippocamp R (*p* < 0.001, q = 0.001), and Precuneus–PCC (*p* = 0.040, q = 0.048). While the comparison of the connectivity strength between CID and HC yielded an uncorrected *p*-value of 0.039 for the PCC–AIR, the FDR correction with adjusted q = 0.058 was nearly significant.

### 3.4. Correlation Between Connectivity Strengths, Questionnaires, and PSG Parameters

Pearson correlation analysis was performed, and the results are presented in Table 4. A significant negative association between DLPFC to Hippocamp R connectivity and ISI scores (r = –0.51, *p* < 0.001) was identified. We also found significant negative correlations between this connectivity and BDI scores (r = –0.27, *p* < 0.05), total wake time (r = –0.32, *p* < 0.05), and WASO (r = –0.30, *p* < 0.05). ISI scores were positively correlated with BDI (r = 0.70, *p* < 0.001), total wake time (r = 0.49, *p* < 0.01), SOL (r = 0.30, *p* < 0.05), and WASO (r = 0.45, *p* < 0.001). All the correlations remained significant after the two-stage FDR correction.

## 4. Discussion

The major findings of this study indicate that there are distinct alterations in the reciprocal connectivity patterns among hubs from the DMN, SN, and ECN in patients with insomnia as compared to healthy individuals. We observed statistically significant altered EC between the patients and the HC group in five connections, namely from DMPFC to VMPFC (excitatory), from MPFC to AIR (excitatory), from PCC to AIR (excitatory), from Precuneus to PCC (excitatory), and from DLPFC to Hippocamp R (inhibitory). Moreover, the coupling values for the DMPFC–VMPFC, MPFC–AIR, and DLPFC–Hippocamp R connections significantly differed from zero only in the patients’ group, whereas the PCC–AIR and Precuneus–PCC connections were significantly different from zero in the HC group (lacking in the patients’ group).

The MPFC has been proven to be a pivotal structure for the pathophysiology of insomnia. Reduction in the regional homogeneity (ReHo) values in MPFC coupled with increased ReHo in the cuneus has been observed in patients with chronic insomnia with and without cognitive impairment, as opposed to HC [36]. Furthermore, several studies have demonstrated a statistically significant improvement of sleep quality after low-frequency repetitive transcranial magnetic stimulation of the DLPFC, which also resulted in alteration of the serum BDNF and GABA levels and of the tissue GABA+/Cr concentration [37,38]. Considering the primary functions of the MPFC related to emotion-processing, motor control, and decision making, we may speculate that the observed excitation of the AIR leads to hyper-stimulated salience, which prevents the individual from falling asleep and maintaining the sleep state [39,40,41]. The increased excitatory bias may also result in disruption of the integrative insular function, leading to disorganization of the “switching” between the DMN and ECN, and the resulting failure to inhibit wakefulness, which is in concordance with the hypothesized hyperarousal.

Another finding that supports the aforementioned impaired “switching” mechanism is the observed excitatory PCC–AIR connection in HC, which is not observed in the CID group. Impaired connectivity between those key nodes of both networks is reported in a study exploring FC in patients with impaired cognitive control, highlighting the role of SN in reducing DMN activity at rest [42]. The authors found that the connectivity strength is increased in their HC group, which is in concordance with our data, and elucidates the importance of these interactions, as it pertains to the attentional detection of salient stimuli and reducing DMN activity when attention is externally focused. Nonetheless, Chen et al. reported overactivation of the insular region in insomniacs [43]. A potential explanation for the observed difference could be found in the methodology of their study, as their testing is performed when the CID patients are instructed to fall asleep during the scan, a fact that implies active engagement of the otherwise passive process of sleep initiation. The latter phenomenon, corresponding to the attention-intention-effort hypothesis of insomnia, could be the cause and not the consequence of the overactivated insula [44,45]. However, it should be noted that the actual role of the SN is more intricate and that there is marked complexity of the possible mechanisms, altering the regulation between the brain networks.

Of particular note is the previously reported right lateralization of the AIR’s ability to modulate and act as a switch mechanism, which is a phenomenon observed in our study as well [35]. This function involves detecting salient internal and/or external stimuli and initiating a shift from self-referential (DMN) to goal-directed (ECN) processing [46]. Our data align with the results of Li et al., suggesting aberrant salience processing, supporting the idea of impaired modulation between internally oriented to externally directed cognitive processing [35]. In contrast to these results, Long et al. report altered insular FC after sleep deprivation, particularly in young adults, suggesting that the impaired signalization between the control networks is a consequence of impaired sleep and not a causative factor [47]. On the one hand, these discrepancies may be attributed to the methodological differences (temporal vs. causative correlations), while on the other hand the alterations observed by Long et al. may reflect short-term compensations, compared to the long-term changes observed in insomniacs, indicating dysfunctional top-down regulation.

Additionally, the excitatory Precuneus–PCC connection, present only in HC, could be responsible for the maintenance of functional integrity within the DMN, and we could speculate that its absence in the CID group reflects poor internal regulation and increased mind-wondering or rumination in patients. Our hypothesis is based on the previously reported data, suggesting that decreased FC between key nodes of the DMN is in the basis of CID and contributes to the inability to disengage from internal thoughts, which may underlie the hyperarousal state [48,49,50].

Furthermore, it should be noted that changes in the gray matter volume and connectivity patterns of the prefrontal cortex and insula have been observed not only in CID but also in major depressive disorder and anxiety disorders [51,52]. These conditions frequently co-occur with CID, which may indicate the existence of overlapping underlying pathogenetic mechanisms, especially pertaining to the impaired emotion regulation and cognitive dysfunction [3]. Moreover, dysconnectivity of the insular cortex has been suggested as a possible underlying factor for ruminations, which are seen in both major depressive disorder and anxiety disorders and can be considered as a potential cause of sleep disturbances [53].

The DLPFC, a major hub of the ECN, has been associated with working-memory capacity, attention regulation, and inhibitory self-control [54,55]. A previous study, assessing brain activations via functional near-infrared spectroscopy, has shown that prefrontal cortex FC is impaired in patients with short-term insomnia disorder, with the connectivity strength being significantly correlated with the severity of the sleep disorder [56]. Additionally, targeting the DLPFC with transcranial magnetic stimulation has shown clinical improvement in CID patients, which persisted after the treatment course [38]. Our findings concerning the inhibitory connection between DLPFC and Hippocamp R could provide further insight into the importance of those regions in CID. We speculate that this could lead to impaired emotional regulation and consolidation of episodic memory. Both phenomena are common in insomniacs, potentially leading to fragmented sleep and thought ruminations [57,58,59]. Likewise, an overly activated DLPFC may trigger excessive top-down control, reflecting the cognitive hyperarousal state [60]. On the other hand, this hypothesis is not consistent with previous findings, which describe reduced metabolism and activity of the prefrontal cortex in insomnia patients at rest; this underscores the complexity of this hypothesis, suggesting that additional factors may also account for the hyperarousal in CID [61].

Additionally, the DLPFC to Hippocamp R connection in our study shows a significant negative correlation with the ISI and BDI scores, indicating that the stronger the inhibition is, the higher the severity of the insomnia and altered emotion regulation. Similar findings are reported by authors exploring insomnia severity in relation to the level of prefrontal activation [62,63]. These results emphasize the contribution of DLPFC to the mechanisms responsible for the regulation and maintenance of normal sleep.

Our findings contribute to the understanding of CID as a network-level disorder rather than a dysfunction of separate neural regions. Altered EC in key hubs like the MPFC, DLPFC, and anterior insula may serve as potential biomarkers for disease severity, treatment response, and prognosis. Future longitudinal studies, including those employing interventional procedures (e.g., rTMS targeting the DLPFC), could evaluate whether normalizing these connectivity patterns results in symptom improvement. Additionally, larger samples and multimodal imaging (e.g., EEG-fMRI fusion) may elucidate whether these EC alterations precede symptom onset, which could inform early identification and prevention strategies, or develop as a result of altered neural connections.

To the best of our knowledge, this is the first study to explore EC across chronic insomnia patients and HC, utilizing dynamic causal modeling analysis. While this study provides valuable insights into the altered connectivity patterns, which elucidate the potential neurologic mechanisms underlying the hyperarousal model for insomnia, there are some limitations, which warrant consideration. First, the sample size is relatively small, which may limit the interpretability of the results and the extrapolation of data. Second, potential confounding factors, such as duration of the insomnia symptoms or variability of the sleep quality for the period during which the patients underwent scanning, were not exclusively controlled. Furthermore, as this is a cross-sectional study, we cannot directly identify causal relations or temporal progression of connectivity patterns. Future research should address those limitations by increasing the sample size, controlling for possible confounding factors and enhancing translational relevance.

## 5. Conclusions

The findings of this study shed light on the potential neurologic substrates of the hyperarousal model of CID. Our results indicate that EC among the DMN, ECN, and SN is altered in individuals with CID. In particular, disruptions within the DMN and between the DMN, SN, and ECN reflect an impaired ability to appropriately shift between internally and externally directed cognitive states—an imbalance potentially underlying the hyperarousal state of CID.

## Figures and Tables

**Figure 1 biomedicines-13-01293-f001:**
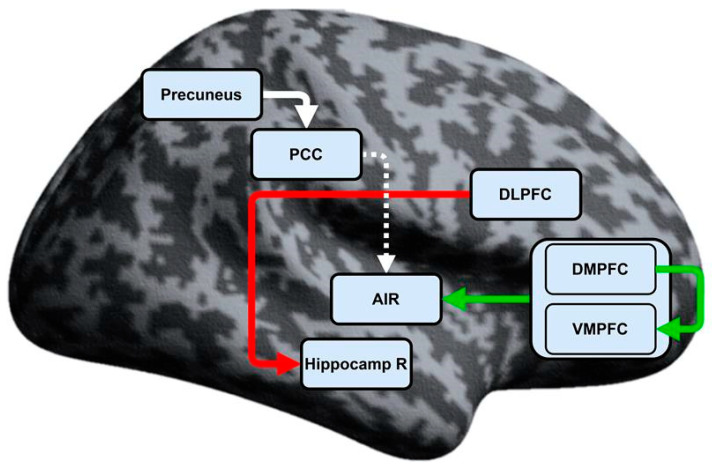
Resting-state effective connectivity in chronic insomnia disorder. Legend: DMPFC—Dorsomedial prefrontal cortex; VMPFC—Ventromedial prefrontal cortex; AIR—Right anterior insula; DLPFC—Dorsolateral prefrontal cortex; Hippocamp R—Right hippocampus; PCC—Posterior cingulate cortex. The common MPFC is displayed as the merged areas of DMPFC and VMPFC. For exact MNI coordinates, see Table 1. The green arrow indicates excitatory connections, present only in the patients’ group. Red arrow indicates inhibitory connections, present only in the patients’ group. White arrow indicates significant connections, lacking in the patients’ group (present only in the HC group). Dashed arrow—False discovery rate correction near significant (q = 0.058).

**Table 1 biomedicines-13-01293-t001:** Regions of interest with their Montreal Neurological Institute (MNI) coordinates.

Region of Interest	X	Y	Z	Brodmann Area
Ventromedial Prefrontal Cortex (VMPFC)	44	52	−2	25
Dorsomedial Prefrontal Cortex (DMPFC)	4	30	46	32
Medial Prefrontal Cortex (MPFC)	3	54	−2	14;11
Right Anterior Insula (AIR)	38	22	3	13
Right Hippocampus	24	−12	−20	28
Dorsolateral Prefrontal Cortex (DLPFC)	−37	27	44	46
Posterior Cingulate Cortex (PCC)	0	−52	26	23;31
Precuneus	−10	−64	24	7

**Table 2 biomedicines-13-01293-t002:** Demographic and clinical data.

Variable	Insomnia (n = 31) Mean (±SD)	Healthy Control (n = 24) Mean (SD)	Between-Group Differences *p*-Value ^ϯ^
Age	34.00 (±9.08)	30 (±7.01)	0.092
Sex (F/M)	22/9	15/9	Χ^2^ = 0.579
Education (secondary/higher)	10/21	6/19	Χ^2^ = 0.496
ISI	18 (±4.00)	4 (±2.85)	<0.001 **
BDI	13 (±7.57)	5 (±4.72)	<0.001 **
ESS	4 (±3.72)	6 (±3.52)	0.040 *
SOL (min)	18 (±15.42)	12.9 (±8.29)	0.325
TIB (min)	474.8 (±60.01)	411.6 (±55.81)	<0.001 **
TST (min)	393.2 (±65.73)	368.8 (±53.42)	0.105
SE	82.3 (±10.65)	79.9 (±27.34)	0.062
WASO (min)	63.6 (±37.08)	30.9 (±17.19)	<0.001 **
N1 (min)	5.6 (±4.37)	4.8 (±4.31)	0.216
N1 (%)	1.4 (±1.17)	1.3 (±1.21)	0.312
N2 (min)	222.7 (±50.08)	183.0 (±51.29)	0.003 *
N2 (%)	56.9 (±9.68)	49.3 (±9.68)	0.010 *
N3 (min)	91.8 (±39.11)	110.8 (±31.40)	0.047 *
N3 (%)	23.2 (±9.73)	30.7 (±9.57)	0.007 *
REM (min)	72.3 (±28.29)	70.2 (±26.23)	0.882
REM (%)	18.1 (±4.78)	18.7 (±5.90)	0.465
REM Latency	115.7 (±61.86)	95.0 (±42.35)	0.243
AHI	2.6 (±2.29)	3.1 (±2.51)	0.191
ODI	1.8 (±1.67)	2.5 (±3.0)	0.347
PLMS Index	3.4 (±5.46)	6.3 (±9.65)	0.176

Legend: ISI—Insomnia severity index; BDI—Beck depression inventory; ESS—Epworth sleepiness scale; SOL—Sleep onset latency; TIB—Time in bed; TST—Total sleep time; SE—Sleep efficiency; WASO—Wake after sleep onset; N1—Stage 1 NREM Sleep; N2—Stage 2—NREM Sleep; N3—Stage 3 NREM Sleep; REM—Rapid eye movement; AHI—Apnea-hypopnea index; ODI—Oxygen desaturation index; PLMS—Periodic limb movement syndrome; ^ϯ^—Kolmogorov–Smirnov test; Χ^2^—Chi-square; * *p* < 0.05; ** *p* < 0.001.

**Table 3 biomedicines-13-01293-t003:** Connections demonstrating significant differences between both groups.

Connections	CID Mean ± SD	HC Mean ± SD	Significance ^U^
DMPFC → VMPFC	0.20608 ±0.342	0.35446 ^a^ ±0.369	0.049
MPFC → AIR	0.06669 ±0.172	−0.06238 ^a^ ±0.221	0.014
DLPFC → Hippocamp R	−0.11002 ±0.258	0.18370 ^a^ ±0.204	<0.001
PCC → AIR ^b^	−0.01701 ^a^ ±0.269	0.14839 ±0.242	0.039
Precuneus → PCC	0.30175 ^a^ ±0.302	0.44117 ±0.388	0.040

Legend: CID—Chronic insomnia patients; HC—Healthy controls; ^U^—Between-group independent samples Mann–Whitney test; ^a^—Not significantly different from 0; ^b^—False discovery rate correction nearly significant; DMPFC—Dorsomedial prefrontal cortex; VMPFC—Ventromedial prefrontal cortex; MPFC—Medial prefrontal cortex; AIR—Right anterior insula; DLPFC—Dorsolateral prefrontal cortex; Hippocamp R—Right hippocampus; AIR—Right anterior insula; PCC—Posterior cingulate cortex.

**Table 4 biomedicines-13-01293-t004:** Correlation analysis.

Correlation	Significance ^r^
DLPFC → Hippocamp R–ISI	−0.507 **
DLPFC → Hippocamp R–BDI	−0.272 *
DLPFC → Hippocamp R–Total wake time	−0.320 *
DLPFC → Hippocamp R–WASO	−0.304 *
ISI-BDI	0.703 **
ISI–Total wake time	0.494 **
ISI-SOL	0.297 *
ISI-WASO	0.448 **

Legend: ^r^—Pearson correlation coefficient; * *p* < 0.05; ** *p* < 0.001; DLPFC—Dorsolateral prefrontal cortex; Hippocamp R—Right hippocampus; ISI—Insomnia severity index; BDI—Beck depression inventory; WASO—Wake after sleep onset; SOL—Sleep onset latency.

## Data Availability

All the data are available from the authors upon reasonable request.

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
