# Peer review of "Aberrant Effective Connectivity Within and Between the Default Mode, Executive Control, and Salience Networks in Chronic Insomnia Disorder—Toward Identifying the Hyperarousal State"

_biomedicines, 2025, doi:10.3390/biomedicines13061293_

Round 1
Reviewer 1 Report
Comments and Suggestions for Authors
Aberrant Effective Connectivity Within and Between the Default Mode, Executive Control, and Salience Networks in Chronic Insomnia Disorder – toward identifying the hyperarousal state
I read the manuscript with interest, and you can find my appraisal, concerns, and suggestions as follows:
Introduction: The section is well-written. However, in line 44, “implicating dysregulation of neural circuits” seems to be quite vague, and you need to be more specific. Indeed, it is not clear if you mean an anatomical or biochemical alteration of these circuits. Please, be more specific. Moreover, as you stated, the resting-state, which is a widely used paradigm in fMRI, has some aspects that are similar to sleep. This statement needs to be taken cautiously since in several previously published studies, during the rs, the participants (or patients) are asked to fix a cross depicted on a monitor. Please, explain this sentence better. DMN coherence is usually used for EEG studies. Please clarify what you mean by coherence and EC. The resting state networks that you introduced need to be described in a better way. Please add more information about the anatomical systems involved in each network. The hypotheses are interesting, but they need to be stated better. Moreover, you need to highlight the hyperarousal theory that was only mentioned, and you used it as the frame of reference.
Methods: The section is well-written, and the methods used are explained in a detailed way. However, it is not clear the use of the 8 ROIs. This needs to be justified in the introduction or at least in the present section.
Results: In this section, you need to describe the connectivity results better. But I like the tables and the figure. I suppose that the reported results are corrected, but it is not specified. This needs to be specified. Moreover, did you correct for multiple comparisons the results of the correlation?
Discussion: The section is interesting, but I advise expanding the future direction and adding more about the clinical impact of your findings.
Author Response
Dear Reviewer, we appreciate the time and effort You have dedicated to providing feedback on our paper “Aberrant Effective Connectivity Within and Between the Default Mode, Executive Control and Salience Networks in Chronic Insomnia Disorder – toward identifying the hyperarousal state”. Please see the attachment for the detailed, point-by-point response to your comments.

Reviewer 2 Report
Comments and Suggestions for Authors
1-)results can be more specific.
here were three 21
significant connections, present in the CID group – inhibitory from the dorsolateral pre- 22
frontal cortex (DLPFC) to the right hippocampus (Hippocamp R), excitatory from the dor- 23
somedial prefrontal cortex to the ventromedial prefrontal cortex, and excitatory from the 24
common medial prefrontal cortex to the right anterior insula (AIR).
2-)authors can add more keywords.
3-)the figure 1 maybe not very clear.
why red and green colors are used?
4-)the discussion part can be improved highlighting the novelty of the study.
5-)authors can add more references for the introduction section.
Author Response

(The authors gave the same response as above.)

Round 2
Reviewer 1 Report
Comments and Suggestions for Authors
The authors have addressed all the concerns that I have raised.
However, in line 78, I suggest " altered Intrinsic connectivity in DMN" Instead of "disrupted synchronization within the DMN". It is a suggestion.